# Multi-Omics Profiling of *Candida albicans* Grown on Solid Versus Liquid Media

**DOI:** 10.3390/microorganisms11122831

**Published:** 2023-11-22

**Authors:** Rouba Abdulsalam Alhameed, Mohammad H. Semreen, Mohamad Hamad, Alexander D. Giddey, Ashna Sulaiman, Mohammad T. Al Bataineh, Hamza M. Al-Hroub, Yasser Bustanji, Karem H. Alzoubi, Nelson C. Soares

**Affiliations:** 1Research Institute of Medical and Health Sciences, University of Sharjah, Sharjah P.O. Box 27227, United Arab Emirates; rouba.a.alhameed@gmail.com (R.A.A.); mhamad@sharjah.ac.ae (M.H.); ashnasulaiman19@gmail.com (A.S.); halhroub@sharjah.ac.ae (H.M.A.-H.); ybustanji@sharjah.ac.ae (Y.B.); kelzubi@sharjah.ac.ae (K.H.A.); 2Department of Medicinal Chemistry, College of Pharmacy, University of Sharjah, Sharjah P.O. Box 27227, United Arab Emirates; 3College of Health Sciences, University of Sharjah, Sharjah P.O. Box 27227, United Arab Emirates; 4Center for Applied and Translational Genomics, Mohammed Bin Rashid University of Medicine and Health Sciences, Dubai P.O. Box 505055, United Arab Emirates; alexander.giddey@mbru.ac.ae; 5Center for Biotechnology, Department of Molecular Biology and Genetics, College of Medicine and Health Sciences, Khalifa University, Abu Dhabi P.O. Box 127788, United Arab Emirates; mohammad.bataineh@ku.ac.ae; 6College of Medicine, University of Sharjah, Sharjah P.O. Box 27227, United Arab Emirates; 7School of Pharmacy, The University of Jordan, Amman 11942, Jordan; 8Department of Pharmacy Practice and Pharmacotherapeutics, College of Pharmacy, University of Sharjah, Sharjah P.O. Box 27227, United Arab Emirates; 9Laboratory of Proteomics, Department of Human Genetics, National Institute of Health Doutor Ricardo Jorge (INSA), 1649-016 Lisbon, Portugal; 10Centre for Toxicogenomics and Human Health (ToxOmics), Faculdade de Lisboa, NOVA School, 1169-056 Lisbon, Portugal

**Keywords:** *Candida albicans*, proteomics, metabolomics, LC-MS/MS, environmental adaption

## Abstract

*Candida albicans* is a common pathogenic fungus that presents a challenge to healthcare facilities. It can switch between a yeast cell form that diffuses through the bloodstream to colonize internal organs and a filamentous form that penetrates host mucosa. Understanding the pathogen’s strategies for environmental adaptation and, ultimately, survival, is crucial. As a complementary study, herein, a multi-omics analysis was performed using high-resolution timsTOF MS to compare the proteomes and metabolomes of Wild Type (WT) *Candida albicans* (strain DK318) grown on agar plates versus liquid media. Proteomic analysis revealed a total of 1793 proteins and 15,013 peptides. Out of the 1403 identified proteins, 313 proteins were significantly differentially abundant with a *p*-value < 0.05. Of these, 156 and 157 proteins were significantly increased in liquid and solid media, respectively. Metabolomics analysis identified 192 metabolites in total. The majority (42/48) of the significantly altered metabolites (*p*-value 0.05 FDR, FC 1.5), mainly amino acids, were significantly higher in solid media, while only 2 metabolites were significantly higher in liquid media. The combined multi-omics analysis provides insight into adaptative morphological changes supporting *Candida albicans*’ life cycle and identifies crucial virulence factors during biofilm formation and bloodstream infection.

## 1. Introduction

### 1.1. Background Information

*Candida albicans* is the most prevalent pathogenic fungus among the *Candida* species and the primary source of fungal infections worldwide [1,2,3,4]. In immunocompetent individuals, *C. albicans* is considered a harmless commensal fungus, whereas in immunocompromised individuals, *C. albicans* infections may lead to invasive mycoses or, in more serious cases, bloodstream infections [5,6,7]. Recently, candidiasis has been increasingly associated with high morbidity/mortality rates and thereby represents a heavy burden to healthcare systems worldwide [8]. *C. albicans* is well known for its remarkable morphological plasticity, which enables a reversible transition between the unicellular yeast form and filamentous form (hyphae and pseudohyphae). This morphological plasticity is one of the most essential pathogenic and virulent characteristics of this microorganism [9,10,11,12,13]. For example, in the yeast form, *C. albicans* can adhere to endothelial cells (the first line of defense that line the inner part of blood vessels) and migrate into the bloodstream, whereas the hyphae form is essential for tissue penetration and invasion [14,15,16,17,18,19]. The yeast-to-hyphae switching/transition is triggered by external stimuli, including nutrient deficiency, elevated temperature (37 °C), low levels of oxygen, and high levels of CO2 [20,21,22]. Other virulence factors include invasion and secretion of cytolytic toxin [7,23,24,25]. Among the most significant virulence factors is biofilm formation, which is a structured microbial community that can be formed on living (host cell) and non-living (abiotic) surfaces and is embedded in an extracellular matrix [26,27]. Clinically, when biofilms are formed on human tissue or on an implanted medical device, they become a main source of drug resistance and infection, spreading to other body parts [26,28]. *C. albicans’* life cycle includes those physiological transformations seen during the production of biofilms, surface adhesion, and host–pathogen interactions [21,22,29]. Therefore, to improve the current approaches in the prevention and treatment of *C. albicans* infections, we must gain insights into the molecular mechanisms mediating these processes of environmental adaptations, survival, and, ultimately, pathogenicity.

### 1.2. Multi-Omics Approach in Microorganisms

Mass spectrometry has significantly improved microbial proteomics, enabling the identification of membrane, cellular, periplasmic, and extracellular proteins, as well as their expression and posttranslational modifications [30]. Shotgun proteomics is the simplest method used in bottom-up methodologies. Usually, a reverse phase C18 column is used to fractionate this peptide mixture before being subjected to mass spectrometric analysis, and it is used in conjunction with other separation techniques [31,32]. Metabolites are small molecules, typically with a low molecular weight between 50 and 1500 Da. Metabolites take part in metabolic processes that are essential for cellular growth, maintenance, and functions, and they respond to persistent alterations in the environment [33,34].

Recently, multi-omics analysis has made a major contribution to the field of medical microbiology [35,36,37,38], specifically liquid chromatography-mass spectrometry-based, and proteomics/metabolomics approaches have provided the research community with key molecular and cellular elements that provide a comprehensive understanding of microbial system biology and important aspects of microbial physiology [35,36,39]. Of note, until now, most of the multi-omics studies performed on *C. albicans* were performed in liquid media, which has been, beyond any doubt, a valuable contribution to this research field (e.g., [40,41,42,43,44]). However, there are well-known limitations in performing proteomics/metabolomics analysis in liquid medium. For example, in these circumstances, microbes experience significant osmotic pressure, which causes the expression of genes related to cell adhesion and biofilm formation to be among the most downregulated [45,46]. This explains why earlier omics studies employing liquid media were unable to identify some of the crucial molecular determinants related to virulence, cell adhesion, and/or responses to specific stresses such as desiccation and aerobic stress. Within this context, recently, we have demonstrated that pathogenic bacteria display distinctive proteome/metabolome signatures in bacterial cells maintained on agar plate solid medium [47,48,49,50]. In this line of research, this study uses Liquid Chromatography-Tandem Mass Spectrometry (LC-MS/MS) based high throughput technology as a novel complementary strategy to evaluate the proteome and metabolome of *C. albicans* grown on agar plates versus suspension liquid culture to shed light on the various environmental adaptations, morphological transformation, and survival.

## 2. Materials and Methods

### 2.1. Solvents and Reagents

Pierce IP Lysis buffer (250 mL) was purchased from Thermo Scientific, Rockford, IL, USA. Methanol (≥99.9%), chloroform, acetonitrile, LC-MS CHROMASOLV, and deionized water were purchased from Honeywell (Wunstorfer Strasse, Seelze, Germany). Pierce trypsin protease, lysyl-endopeptidase LysC, and C18 columns were purchased from Thermo Scientific (Rockford, IL, USA). Formic acid (FA) was purchased from Fisher Chemical (Geel, Belgium). Trifluoroacetic acid (TFA) was purchased from Fisher Scientific (Loughborough, UK). Hydrochloric acid (HCl) (37%) was purchased from VWR Chemicals (Rosny-sous-Bois, France). Bovine serum albumin and Bradford’s reagent were purchased from Sigma-Aldrich (St. Louis, MO, USA).

### 2.2. Yeast Sample Preparation

Wild Type (WT) *Candida albicans* DK318 [51] was used as the study material in this experiment. Yeast potato dextrose (YPD) (HiMedia, Thane, India) broth and solid culture media were used for culturing *C. albicans*. First, pure *Candida albicans* colonies were isolated from the glycerol stock streaked on YPD agar and incubated for 24 h. For the broth culture, two to three isolated *C. albicans* colonies were inoculated in 100 mL of YPD agar and incubated at 37 °C while shaking at 200 rpm for 24 h. After incubation, cells from the broth culture were harvested with centrifugation at 4000 rpm for 10 min in a pre-weighed tube. Later, the cell pellets were washed with sterile phosphate buffer saline (PBS) to remove the culture media and then air-dried to measure the biomass. The cultures at the time of harvest were in the stationary phase. For agar culture, fresh YPD agar plates were streaked using the isolated colonies of *C. albicans* and incubated at 37 °C for 24 h. Yeast biomass from the agar plates was collected using a sterile loop, washed with PBS, and pelleted in a pre-weighed sterile conical tube. Pellets were then air-dried to measure the biomass. 

### 2.3. Proteomics and Metabolomics Extraction

A protocol recently developed in our lab [52] was used to extract both proteins and metabolites from each replicate pellet. Briefly, 400 µL of IP lysis buffer containing one tablet of protease inhibitor per 10 mL of lysis buffer (Pierce, Thermo Scientific) was added to each replicate. The pellets were homogenized by vortexing for 30 s and incubated for 10 min at room temperature prior to three rounds (30 s each) of sonication using a COPLEY sonicator (Qsonica, Newtown, CT, USA) at 30% amplitude. Sample lysates were transferred to 1.5 mL microcentrifuge tubes, centrifuged at 14,000 rpm for 5 min, and the supernatants containing the proteins and metabolites were transferred to fresh microcentrifuge tubes. Proteins and metabolites were separated through protein precipitation: 400 µL of methanol was added to each replicate, followed by 300 µL of chloroform, then vortexed. Samples were centrifuged at 14,000 rpm for 5 min to obtain a biphasic solution with proteinaceous interphase and polar and non-polar metabolites in the upper and lower phases, respectively. For each sample, the upper supernatant containing the (polar) metabolites was transferred carefully to a new microcentrifuge tube without disturbing the white disk. The interphase and the lower phase were mixed with 300 µL of methanol, vortexed, and then centrifuged for 5 min at 14,000 rpm to pellet the protein. The remaining (non-polar) supernatant was added to that previously collected. The combined metabolite extractions were entirely dried using EZ-2 Plus (GeneVac, Ipswich, UK) at 45 °C, and samples were reconstituted in 200 µL of 0.1% formic acid in LC-MS grade water. Finally, samples were filtered using 0.22 µm filters and transferred into a micro-insert to be analyzed using Trapped Ion Mobility Spectrometry-Time-of-Flight-MS (TIMS-TOF MS). As for the proteins, after removing the metabolite layer, the protein pellets were air dried at room temperature and then resuspended in 100 µL denaturation buffer (6 M urea, 2 M thiourea in 10 mM Tris buffer, pH 8). Protein quantitation was determined using a modified Bradford assay [53] before the tryptic protein digestion (see below). 

### 2.4. In-Solution Protein Tryptic Digestion and Desalting

Protein samples (100 µg per sample) were reduced by the addition of dithiothreitol (DTT, 1 mM final concentration) and incubated for 1 h at 100 rpm at room temperature (RT). Reduced disulfide bonds were capped by alkylation with the addition of iodoacetamide (IAA, final concentration of 5.5 mM) and incubated for 1 h at 100 rpm in the dark at RT. After the incubation, the pH was measured and ensured to be equal to 8. One µg of lysyl endopeptidase LysC enzyme was added to samples and incubated for 3 h at 100 rpm, RT. Afterward, samples were diluted with four volumes of 20 mM ammonium bicarbonate before the addition of 1 μg of trypsin, and then samples were incubated overnight at RT. Finally, samples were completely dried under vacuum using an EZ-2 Plus (GeneVac, Ipswich, UK) before desalting and sample clean-up.

The STop And Go Extraction (STAGE) method was used for sample desalting. In brief, samples were resuspended in 100 µL of 1% trifluoroacetic acid (TFA). The pre-conditioned C18 STAGE tips with bound proteins were then rinsed twice with 100 µL of 0.1% TFA in 5% acetonitrile (ACN). Samples were eluted using 0.1% formic acid (FA) in 60% ACN, then dried completely using an EZ-2 Plus and resuspended in 100 µL of 0.1% FA in 2% ACN prior to LC-MS/MS analysis.

### 2.5. Ultra High-Performance Liquid Chromatography-Tandem Mass Spectrometry (UHPLC–MS/MS) Nano-Proteomics

The LC-MS/MS analysis for sample separation was performed using a nano ELUTE (Bruker Daltonics, Billerica, MA, USA) coupled to a quadrupole time-of-flight mass spectrometer (Q-TOF) utilized with a Captive Spray ion source. Four micrograms of the peptides/sample were injected and separated using 15 C18 columns (15 cm × 75 μm, 1.9 μm, Bruker, Billerica, MA, USA), and the Captive Spray ion source conditions for every injection were as follows: the drying gas flow rate was 3.0 L/min at a temperature of 150 °C, and the capillary voltage was set at 1600 V. The elution gradient over 140 min using solvent A (0.1% FA in deionized water) and solvent B (0.1% FA in ACN) was as follows: 0 to 5 min, 5% B; 5 to 120 min, 5–35% B; 120 to 125 min, 35–95% B; 125 to 135 min, 95% B; 135 to 135.2 min, 95–5% B; 135.2–140, 5% B. The acquisition was performed using the positive mode at 2 Hz, the scan range was from 150 to 2200 *m*/*z*, the width of the precursor ion was ±0.5, the cycle time was 3.0 sec, and the threshold was 500 counts. Within a 2-min window, active exclusion was introduced. As a function of the m/z parameter, the collision energy was adjusted between 23 and 65 eV.

### 2.6. Ultra-High-Performance Liquid Chromatography Coupled to Electrospray Ionization and Quadrupole Time-of-Flight Mass Spectrometry (UHPLC-ESI-QTOF-MS) Metabolomics

The LC-MS/MS analysis for metabolomics was performed using an ultra-high-performance liquid chromatography system, the Elute UHPLC (Bruker Daltonik GmbH, Bremen, Germany) coupled to a quadrupole time-of-flight mass spectrometer (Q-TOF). A 10 microliter aliquot of the sample was injected and separated using an Intensity Solo C18 column (2.1 mm × 100 mm, 1.8 µm) (Bruker Daltonik) at a column oven temperature set at 35 °C, using solvent A (0.1% FA in deionized water) and solvent B (0.1% FA in ACN) with the following gradient elution mode: 0 to 2 min, 1% B; 2 to 17 min, 1–99% B; 17 to 20 min, 99% B; 20 to 20.1 min, 99–1% B; 20.1 to 30 min, 1% B. The flow rate was 0.25 mL/min from 0 to 20 min, 0.35 mL/min from 20 min to 28.3 min, and 0.25 mL/min from 28.3 to 30 min. The ESI source conditions for every injection were as follows: the drying gas flow rate was 10.0 L/min at a temp of 220 °C; the capillary voltage was set at 4500 V; the end plate offset was set at 500 V; the nebulizer pressure was 2.2 bar. Auto MS scan acquisition ranged from 0 to 0.3 min for the calibrant sodium formate, and auto MS/MS scan with collision-induced dissociation (CID) acquisition included fragmentation and ranged from 0.3 to 30 min, and was performed using the positive mode at 12 Hz. The automatic in-run mass scan range was from 20 to 1300 m/z, the width of the precursor ion was ±0.5, the cycle time was 0.5 s, and the threshold was 400 counts. After three spectra, active exclusion was excluded and released after 0.2 min. Quality control (QC) samples were prepared by pooling the same volume of each sample, and the injection was performed every 9–10 samples to evaluate the reproducibility of the analysis. 

### 2.7. Bioinformatics Analysis and Statistical Approach 

Omics raw data were processed using MaxQuant and Perseus software (https://maxquant.net/perseus/, accessed on 26 February 2023) for proteomics, and MetaboScape and Metaboanalyst for metabolomics. The peptide spectral matching and protein inference were performed with MaxQuant 2.1.0.0 [54,55] using the reference proteome for *Candida albicans* in the Uniprot human databases (uniprot-proteome-canonical_UP000000559_2022.06.28. fasta). Default parameters were used, except that 1 unique and 2 total peptides per protein were required for protein inference and ‘match-between-runs’ was enabled. Using Perseus software for proteomics analysis, raw data were filtered for only those proteins with at least three valid values in at least one group. Data imputation was performed as per the Perseus default by replacing missing values from a downshifted normal distribution (width: 0.3, downshift: 1.8). Processed metabolite features were matched to the human metabolome database (HMDB) Mass Spectral Library by way of MS^2^ spectra in MetaboScape. 

For both datasets, significance was determined with classical statistical testing using Student’s *t*-test to identify analytes that significantly differed between the two growth conditions (*p* < 0.05) after multiple testing corrections (false discovery rate (FDR)). For visualization with volcano plots using label-free quantification (LFQ) intensities, a log_2_ (fold-change) threshold of >1 in either direction was additionally required to identify the strongly altered analytes.

## 3. Results

The proteome of *C. albicans* has been extensively described elsewhere [40,41,42,43,44]. To our knowledge, there have not been any direct comparisons made between the proteome or metabolome of *C. albicans* (or other pathogenic fungi) in liquid cell culture versus solid media. Herein, this study compares the proteomes and metabolomes of WT *C. albicans* (strain DK318) when grown in solid (agar plates) versus liquid (broth) media, with a specific focus on the changes occurring in solid media (agar plates). Although debatable, the surrounding conditions at the stationary phase are believed to be more like those observed in solid media [56,57], such as nutrient limitations, individual competition, accumulation of byproducts, and, eventually, a steady growth rate. As proof of concept, we selected a single time point 16 h after inoculation for both the solid and liquid stationary phase. See the detailed experimental design (Figure 1).

### 3.1. Proteomics Analysis Reveals That C. albicans Employs Different Protein Machinery to Sustain Growth on Agar Solid and Liquid Media Culture 

Proteomic analysis of protein extracts from solid medium together with suspension liquid medium identified a total of 1793 proteins and 15,013 peptides. Of these, a total of 1403 proteins had valid quantitative values for all three replicates in at least one group (See Materials and Methods Section 2.7). Principal component analysis (PCA) indicated that the sample proteome profiles were separated primarily based on growth media along the first principal component (accounting for ~49.2%), which strongly suggested that these two sample groups (liquid vs. solid growth) presented distinct proteome profiles (See Figure 2A). As a side note, the larger intragroup variation observable in the solid media growth samples on the PCA would seem to corroborate the expected increased heterogeneity in solid growth samples relative to the highly homogenous liquid media, as nutrient access in solid media can differ independently for separate colonies, whereas this would be expected to be equal at any point in time for all organisms in the liquid culture medium. 

The PCA analysis prompted downstream statistical analysis by way of multiple testing-corrected (MTC)-adjusted Student’s *t*-test using label-free quantitative (LFQ) values to pinpoint the significant differences between the differentially deregulated proteins (See Materials and Methods Section 2.7). Out of the 1403 identified proteins, 313 proteins were considered differentially significant deregulated proteins with a *p*-value < 0.05. Of these, 156 proteins significantly increased in suspension liquid medium compared to solid agar plates, whereas 157 proteins decreased (*p*-value < 0.05) in suspension liquid medium compared to solid agar plates. The volcano plot indicated 98 differential proteins with a log2-transformed fold change > 1 in either direction (See Figure 2B), and the gene set enrichment analysis (GSEA) of significant deregulated proteins enriched for terms such as “translation”, “structural constituent of ribosome”, “cytosolic large ribosomal subunit”, and “RNA binding” seemed to be the ones highly impacted when *C. albicans* was grown in liquid compared to solid media (see Figure 2C). Additionally, the analysis (GSEA) retrieved terms “cell surface”, “extracellular region”, and “fungal-type cell wall” were impacted. In comparison with liquid media, cells on solid agar plates surroundings experience a rather slower/static microbial cell growth, and individual colonies are subjected to challenging nutrient availability, as well as several other environmental stressors (pH, temperature, desiccation, oxidative stress, etc.) [47,48,49,50]. In *Candida*, like other microbes, such conditions are known to induce the expression of several virulence factors [58,59], and in concordance, among the differential higher abundant proteins, there are several known virulence factors, as summarized in Table 1-Part I and Part II and discussed later in the text.

### 3.2. Metabolomics Analysis Indicates That C. albicans Possesses a Versatile and Robust Metabolism

The untargeted metabolomics analysis identified 192 metabolites in total. Like the proteomics analysis (see Section 3.1), the two-component analysis showed that *C. albicans’* growth on a solid agar plate or in suspension liquid media produced two metabolome profiles (Figure 3A). The subsequent statistical analysis showed that the abundance of 48 out of 192 metabolites significantly altered between the liquid medium and the solid agar plate (*p*-value 0.05 FDR and fold change 1.5) (Figure 3B). In other words, regardless of the media, 144/192 metabolites remained unchanged (Appendix A). These metabolites most likely form the core of the endogenous *C. albicans* metabolites that are essential for growth and development in any kind of environment; obviously, this warrants further investigation.

Unlike the proteomic data, which indicated an approximately equal number of proteins increased in both solid and liquid media, the metabolomics analysis revealed that a large majority (42/48) of the metabolites, mainly amino acids, were significantly more abundant in agar solid media (Figure 3B and Table 2), whereas only 2 metabolites were significantly higher in suspension liquid media. In contrast, the metabolomics analysis of solid media showed a pronounced increase in the abundance of essential amino acids (e.g., L-proline, L-arginine, L-valine, L-threonine, Aspartame, Homo l-arginine, L-glutamic acid, and others) (see Appendix A and Figure 3C) and fatty acid metabolism (l-Carnitine). Accordingly, among the proteins upregulated in solid media, there were proteins involved in the amino acids metabolism, such as threonine synthase, homoserine O-acetyltransferase, glutaminase, aspartate transaminase, branch-chain-amino-acid-aminotransferase, N-Acetylputrescine (Appendix A), and fatty acids metabolism, for example, carnitine, acylcarnitine antiporter, acly carrier protein, medium-chain fatty acid-CoA ligase, long-chain fatty acid-CoA ligase, long-chain fatty acid transporter, acetyl-coenzyme A synthetase, and acetyl-CoA C-acyltransferase (Appendix A).

## 4. Discussion

Other non-MS-based omics studies have looked at other *Candida* species identification, such as Pezzotti et al.’s, using Raman spectroscopy and imaging for identification and metabolic profiling of *Candida auris* clades and subclades [60]. In another study, Himmelreich et al. utilized nuclear magnetic resonance (NMR) spectroscopy combined with a statistical classification strategy (SCS) to differentiate between *Candida albicans* and *Candida dubliniensis* [61].

Here, we used the complementary MS-based proteomics and metabolomics techniques in this multi-omics study to determine the differences between *C. albicans* grown on solid vs. liquid media. The data presented here indicate that the core growth and replication machinery are enhanced in suspension media, with several terms related to protein translation and core metabolism increased in this media. This is understandable, considering that cells in suspension media are exposed to changing conditions as the population grows, going from an exponential phase to a stationary phase with varying growth rates and different cell morphologies throughout the stages [62,63]. This translates to a continuously changing proteome profile along with changes at both the transcriptional and translational levels [62]. Therefore, it is conceivable that the observed dynamic shift in ribosomal levels and, indeed, at transcription/translation machinery is crucial in ensuring the adaptability of *C. albicans*’ responsive proteome profile (Table 1-Part I and II). These results are, to some extent, understandable, considering that the aqueous environment found in liquid suspension culture facilitates a greater exchange of nutrients, including metabolites and signaling molecules, between the surroundings and the microbes [64,65].

On the other hand, pathogenic cell wall proteins and proteins related to the cell surface were increased in cells grown on solid media. Specifically, cell surface mannoprotein MP65, pH-regulated antigen PRA1, pH-responsive protein 1, RBT4 cholesterol binding protein, inhibitor I9 domain-containing protein, predicted GPI-anchored protein 17, and secreted glucosidase SUN41. The cell surface mannoprotein MP65 is important for cell wall integrity and biofilm formation. Silvia et al. conducted an extensive analysis of a *mp65Δ* mutant to determine the function of this protein in maintaining the integrity of cell walls, adhering to epithelial cells, and forming biofilms. The study revealed that in comparison to the wild type, the mp65 mutant exhibited a markedly impaired biofilm formation process. [66] Arne et al. used a screening methodology to determine whether pH-regulated antigen 1 (Pra1) is a molecule that can bind to mouse CD4+ T cells directly in vitro. The results showed that the mouse CD4+ T cells and *Candida albicans’* pH-regulated antigen 1 (Pra1) bind together directly, which was enhanced by extracellular Zn2+ ions. [67].

Equally relevant is the presence of a protein involved in cell morphogenesis (Opaque-phase-specific protein OP4, Msb2p), filamentous/polarized growth (Mlc1p, chitinases 1 and 2, ras-like protein, profilin), and viability (thioredoxin reductase TRR1, copper transport protein CTR1). Importantly, on the other hand, proteomics analysis of liquid media revealed a different set of elevated virulence factors. However, these are closely associated with host interaction (e.g., Adh1p, pH-responsive protein 2, GST2p, V-type proton ATPase subunit B) and bloodstream infection (e.g., blood-induced peptide 1, probable NADPH dehydrogenase, agglutinin-like protein 2). For example, blood-induced peptide 1 is important for the organism’s ability to survive in host blood because it increases tolerance to environmental stresses such as heat, salt, or cycloheximide, which is required for virulence [68]. Agglutinin-like protein 2 belongs to the agglutinin-like sequence (ALS) gene family and is involved in the adhesion of fungal cells to host and abiotic surfaces [69].

Although distant from the conditions found in the host side of infection, the information of the cellular events occurring during liquid suspension and solid agar plate provide important information regarding the life cycle of *C. albicans* and its molecular determinants with important roles in morphogenesis, survival, virulence, and host–pathogen interactions. In concordance with the proteomics findings, in solid media, there was an increase in metabolites associated with *C. albicans* growth (e.g., Phosphonoacetate, N6-Acetyl-L-lysine, asymmetric dimethylarginine, N-Acetylputrescine, 5′-Methylthioadenosine) and biofilm formation (pipecolic acid, guanosine) (Table 2). Additionally, there was an increase in the level of several metabolites related to *Candidas*’ virulence and response to the host immune system, including S-Adenosylhomocysteine, which can serve as both an immunotoxin and a metabotoxin. When it accumulates to sufficiently high levels, it impairs, restricts, or even kills immune cells [70]. Additionally, shikimic acid demonstrated anti-inflammatory effects [71], and homogentisic acid confers a survival advantage during host infection in *Bordetella parapertussis* [72].

**Table 1 microorganisms-11-02831-t001:** Summary of the *Candida albicans* deregulated proteins involved in virulence, pathogenicity, and host interaction.

Protein UniProt ID	Protein Name	GO–Biological Process (Source Uniprot)	Physiology and Morphogenesis	References
** *Part I—Proteins with increased abundance in solid agar* **				
**Q56XX2**	Cell surface mannoprotein M65	Cell surface, cell adhesion and biofilm formation, response to starvation, filamentous growth	Surface mannoprotein is required for hyphal morphogenesis, surface adherence, and pathogenicity. It plays an important role during biofilm development and maintenance and acts as a major antigen target of host cell-mediated immune response.	[73]
**P87020**	pH-regulated antigen PRA1	Cell surface, hyphal cell wall, adhesion of symbiont to host, Zin ion binding. Evasion of host immune response	Cell surface protein is involved in the host–parasite interaction during *Candida* infection. With MP65, it represents a major component of the biofilm matrix. It sequesters zinc from host tissue and mediates leukocyte adhesion and migration. As a released protein, it controls host complement attack, assisting the fungus in escaping host surveillance. It decreases complement-mediated adhesion, as well as the uptake of *C. albicans* by human macrophages.	
**Q5AG89**	Thioredoxin reductase TRR1	Cell redox homeostasis, cellular response to oxidative stress, fungal biofilm matrix	Belongs to the class-II pyridine nucleotide-disulfide oxidoreductase family and is a good target for potentially board-spectrum antifungal antibodies.	[74]
**A0A1D8PGF8**	Msb2p	Cell surface, site polarized growth, osmosensor activity, filamentous growth, positive regulation of single-species biofilm formation, signal transduction involved in filamentous growth	Mucin Msb2 regulates the cek1 MAPK pathway. Msb2 shedding occurred differentially in cells grown planktonically or on solid surfaces in the presence of cell wall and osmotic stressors.	[75]
**Q5AB48**	RBT4	Cell surface, extracellular space, cholesterol binding	A secreted protein that acts as a virulence factor during infection, such as in posttraumatic corneal infections. It acts as an important antigen in patients with systemic candidiasis and plays a role in protection against phagocyte attack.	[76,77,78]
**A0A1D8PH78**	Farnesyl pyrophosphate synthase	Farnesyltranstransferase activity	Farnesyl pyrophosphate synthase is part of the second module of the ergosterol biosynthesis pathway that includes the middle steps of the pathway.	[79]
**Q5A470**	Opaque-phase-specific protein OP4		Switches between white and opaque phases.	
**Q5AIB2**	SCW4	Cell surface, fungal-type cell wall, carbohydrate metabolic process, cell wall organization	Supports cell wall assembly and integrity.	[80,81]
**Q5AF37**	Inhibitor I9 domain-containing protein	Secreted protein	Virulence and pathogen interaction— is able to induce cell death in planta. The inhibitor I9 domain was more abundant in secretomes of a wide range of necrotizing fungi relative to biotrophs.	[82]
**P43076**	pH-responsive protein 1	Cell septum, cellular bud membrane, cell adhesion, entry into host	Required for apical cell growth and plays a vital role in morphogenesis. It may be integral to the pathogenic ability of the organism.	[83,84]
**P40953** **P40954**	Chitinase 2 Chitinase 3	Filamentous growth of a population of unicellular organisms in response to starvation	Chitinase is involved in the remodeling of chitin in the fungal cell wall. It plays a role in cell separation.	[85]
**Q59XU5**	Ras-like protein 1	Actin fusion focus, cell cortex of cell tip, filamentous growth of a population of unicellular organism in response to heat	Required for the regulation of both the MAP kinase signaling pathway and cAMP signaling pathway. The activation of these pathways contributes to the pathogenicity of cells. Induction of morphological transition from yeast to the polarized filamentous form.	[86]
**A0A1D8PSE1**	Mlc1p	Cellular bud neck contractile ring	Response to the maintenance of polarized growth.	[87]
**Q5AHA4**	Predicted GPI-anchored protein 17	Virulence	Predicted GPI-anchored protein role during fungal host infection	[88,89,90]
**Q59NP5**	Secreted beta-glucosidase SUN41	Single-species biofilm formation in or on the host organism, adhesion of symbiont to host	Cell surface beta-glucosidase is involved in cytokinesis, cell wall biogenesis, adhesion to host tissue, and biofilm formation. It plays an important role in the host–pathogen interaction.	[91,92]
**Q5A786**	Profilin	Actin polymerization and depolymerization	Binds to actin and affects the structure of the cytoskeleton. At high concentrations, profilin prevents polymerization, whereas it enhances it at low concentrations.	[93,94]
**Q59NP1**	Copper transport protein CTR1	Cooper transport	Required for high-affinity copper transport into the cell. It is induced during biofilm formation and contact with macrophages as well as by alkaline pH via RIM101.	[95]
**A0A1D8PL61**	Midasin	ATP hydrolysis activity, ribosomal large subunit export	A nuclear chaperon required for maturation and nuclear export of pre-60S ribosome subunits. It is essential for ribosome maturation in yeast (Saccharomyces cerevisiae).	[96]
**A0A1D8PGE0**	Dynein light chain	Dynein intermediate chain binding	Acts as one of several non-catalytic accessory components of the cytoplasmic dynein complex. It may play a role in changing or maintaining the spatial distribution of cytoskeletal structures.	
** *Part II—Proteins with increased abundance in liquid media—virulence, pathogenicity, and host interaction* **				
**P0CT51**	Blood-induced peptide 1	Host interaction, virulence factor	Plays an important role in survival in host blood through increased tolerance to stress, such as salt or cycloheximide, which is essential for virulence.	[68]
**A0A1D8PPK1**	Probable NADPH dehydrogenase	Steroid binding, steroid metabolic process	Oxidoreductase binds mammalian estrogens with high affinity.	[97,98]
**A0A1D8PP43**	Adh1p	Biological process involved in interaction with the host, biofilm formation	Promotes *C. albicans* pathogenicity by stimulating oxidative phosphorylation.	[99]
**P0CU38**	Agglutinin-like protein 2	Cell adhesion involved in multi-species biofilm formation	A cell surface adhesion protein that mediates both yeast-to-host tissue adherence and yeast aggregation. Plays an important role in the pathogenesis of *C. albicans* infections.	[100,101,102]
**O13318**	pH-responsive protein 2	Hyphal cell wall, fungal-type cell wall	Required for apical cell growth and plays an essential role in morphogenesis. It may be integral to the pathogenic ability of an organism.	
**Q5AFB4**	GST2p	Cellular response to oxidative stress	Required for nitrogen starvation-induced filamentous growth in *C. albicans.*	[103]
**Q59PT0**	V-type proton ATPase subunit B	Autophagy, vacuolar acidification	Plays an important role in resistance to several stresses, as well as in autophagy and virulence.	[104]
**P10613**	Lanosterol 14-alpha demethylase	Sterol 14-demethylase activity, cell growth mode switching from budding to filamentous	Plays an essential role in the third module ergosterol biosynthesis pathway.	[105,106]
**Q5AHH4**	Small heat shock protein 21	Cellular heat acclimation, cellular response to oxidative stress	A heat shock protein required for pathogenicity. Mediates thermotolerance and adaptation to oxidative stress. Plays a role in the capacity of damaging human-derived endothelial and oral epithelial cells during infections. Potentiates resistance to antifungal drugs as well as resistance to killing by human neutrophils.	[107,108]

**Table 2 microorganisms-11-02831-t002:** Summary of the *Candida albicans* altered metabolites associated with virulence, pathogenicity, and host interaction.

Metabolite Accession Number	Metabolite Name	Role in Physiology and Morphogenesis	References
**HMDB00679**	Homocitrulline	Secondary metabolites or non-essential metabolites that may serve a role as defense or signaling molecules.	
**HMDB06050**	o-Tyrosine	Hydroxy radical biomarker of oxidative damage to protein.	[109]
**HMDB02064**	N-Acetylputrescine	Cellular processes include cell cycle progression and growth produced by the breakdown of amino acids. Putrescine and cadaverine are primarily responsible for the foul odor of putrefying flesh but also contribute to the odor of bad breath and bacterial vaginosis.	[110]
**HMDB0000130**	Homogentisic acid	In *Pseudomonas, aeruginosa* is a component of Pyomelamin, a black-brown negatively charged polymer produced during L-Tyrosine catabolism. *Bordetella parapertussis* confers a survival advantage during host infection.	[72,111]
**HMDB0003070**	Shikimic acid	It is an important biochemical intermediate in plants and microorganisms. It is a compound that, when extracted from anise plants, has autoinflammatory effects. Accumulates *Candida maltosa* in the presence of a growth inhibitor herbicide, glyphosate.	[71,112]
**HMDB0000070**	Pipecolic acid	Biofilm formation of *Enterococcus faecalis* and *C. albicans.*	[113]
**HMDB0000133**	Guanosine	Fungal-associated infection.	[114]
**HMDB0004110**	Phosphonoacetate	Physiologically essential metabolites involved in an organism’s growth, development, or reproduction.	
**HMDB0001539**	Asymmetric dimethylarginine	In *C. albicans* lacking CaHmt1, asymmetric dimethylarginine and omega-monomethylarginine levels are decreased.	[115]
**HMDB0001988**	4-Hydroxycyclohexylcarboxylic acid	Microorganisms possess important biological activities, such as antibacterial, anti-inflammatory, and hypoglycemic effects.	[116]
**HMDB00819**	Normetanephrine	Normetanephrine is a metabolite of norepinephrine created by the action of catechol-O-methyl transferase on norepinephrine. The latter can influence microbial pathogenesis, the growth and production of virulence factors in enterotoxigenic and enterohemorrhagic strains of *Escherichia coli.*	[117]
**HMDB002006**	2,3-Diaminopropionic acid	N3-(4-Methoxyfumaroyl)-l-2,3-Diaminopropionic acid is a strong inhibitor of the essential fungal enzyme glucosamine-6-phosphate.	[118]
**HMDB0000206**	N6-Acetyl-l-lysine	A novel 2-oxoglutarate aminotransferase catalyzing the second step of lysine catabolism, the oxidative transamination of the alpha-group of N6-acetyllysine, was identified in *Candida maltose*. The enzyme was strongly induced in cells grown on L-lysine as the sole carbon source.	[119]
**HMDB0001173**	5′-Methylthioadenosine	It has been shown to influence the regulation of gene expression, proliferation, and apoptosis.	

Overall, the integration of the analyzed data sets (metabolomics and proteomics) supports the idea that access to nutrients is significantly restricted in solid media, forcing a change in the metabolism of the microorganisms by promoting the biosynthesis/accumulation of essential amino acids and the search for alternative sources of energy, such as fatty acids, in place of glucose and/or other carbohydrates readily available at earlier stages of suspension liquid media cell cultures.

Finally, it is important to draw attention to the increased level of metabolites such as niacinamide, umbelliferone, and 2,3-diaminopropionic acid on solid agar plates that have an antagonistic effect on the growth of *Candida albicans* and the formation of biofilms. These apparent contradictory results can be explained considering population control and nutrient competition. Although speculative, it is plausible that under harsh conditions, there is a need to control population growth by releasing metabolites that prevent colonies from overgrowing and individual competition, making this group of metabolites of drug interest; of course, this warrants further research. Thus, combined multi-omics analysis of the data sets of liquid and solid media represents an excellent opportunity to better understand the key cellular events orchestrating the adaptative morphological changes supporting *Candida’s* life cycle and identify important virulence factors operating during biofilm formation and bloodstream infection.

## Figures and Tables

**Figure 1 microorganisms-11-02831-f001:**
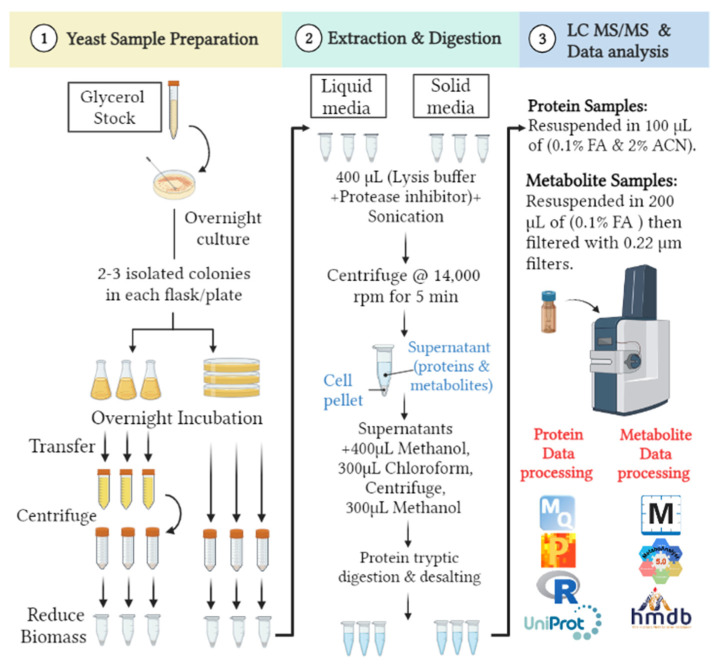
Experimental design comparison between the proteomes and metabolomes of WT *Candida albicans* (strain DK318) when grown with either solid (agar plates) or liquid (broth) media.

**Figure 2 microorganisms-11-02831-f002:**
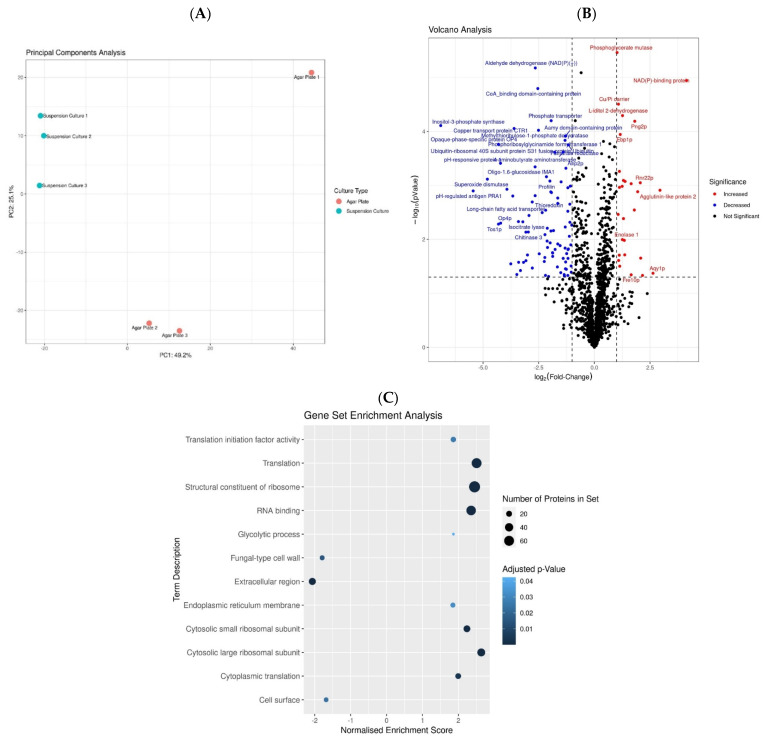
(**A**) Principal component analysis of the proteomics profiles of *C. albicans* from suspension culture and agar plates. (**B**) Volcano plot showing proteins with significantly greater abundance in cells grown in suspension culture (red) or on agar plates (blue). (**C**) Gene set enrichment analysis results. Color corresponds to the enrichment *p*-value, and the *x*-axis is the normalized enrichment score, where positive values indicate increased abundance in suspension culture.

**Figure 3 microorganisms-11-02831-f003:**
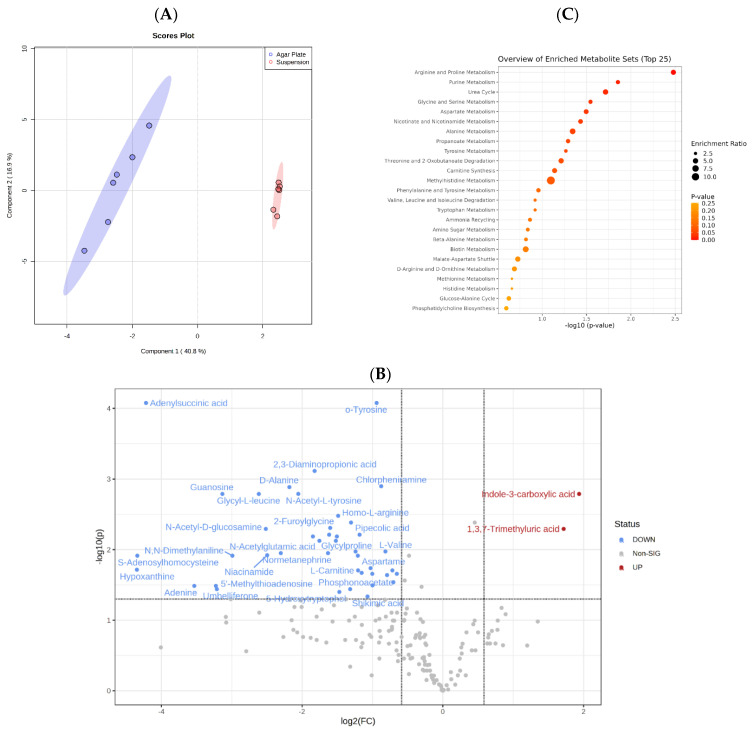
(**A**) Sparse Partial Least Squares Discriminant Analysis (sPLS-DA) shows that the two groups’ metabolic profiles form two distinct clusters. (**B**) Volcano plot displaying the significantly altered metabolites in cells from suspension culture (red color) versus cells grown on agar plates (blue color). Positive log2 (fold change) values correspond with increased abundance in suspension culture. (**C**) Metabolomic shows the most significantly altered metabolic pathways between suspension culture and agar plate cells.

## Data Availability

The proteomic mass spectrometry data have been deposited to the ProteomeXchange Consortium via the PRIDE repository with the data set identifier **PXD04290**, Project name: Multi-Omics profiling of *Candida albicans* from agar plate and suspension media. The metabolomics mass spectrometry data have been deposited to Metabolomics Workbench Consortium with provisory data set identifier Study **ST002730** (datatrack_id: 4070).

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
