# Peer review of "Multi-Omics Profiling of Candida albicans Grown on Solid Versus Liquid Media"

_microorganisms, 2023, doi:10.3390/microorganisms11122831_

Round 1
Reviewer 1 Report
Comments and Suggestions for Authors
Reviewer’s comments to the authors
The following are my comments and critique should be addressed before acceptance:
# As a non-native speaker, the manuscript is easy to read and understand. However, there are many grammatical and punctuation errors and in some instances, phrasing need to improve, and poor punctuation in many sentences. Please ask native speakers to help.
# Albicans in the title should start with a small based on scientific writing roles
# Candida in the abstract should be not bold
# Scientific names should be in italic form
# Define abbreviations at first time that appears in the main text. After the first appearance of the abbreviation, the abbreviation should always be used in the rest of the manuscript instead of the complete term
# plates not plate, (abstract part)
# Make spaces between values and numbers, and also between words
# WT you mean wild type ? needs to be more clear
# isolated or kept ?
# …..incubated for 24 hours….. and how many days?
# RPM in small
# Unify the subheading writing
# Unify units such as minutes/min; hours/h, see all text
# All figures need to be improved (high resolution)
# Title of table 1 should be not in bold
# unify the font of the manuscript (see tables font)
# in tables add a separate column to add references individually
# Insufficient discussion section and the authors should explain more explanation
# Improve and avoid references from the conclusion part
# Recent references should be added
# The authors failed to make a well comparative explanation between the metabolites and proteins produced from C. albicans under various cultivation
Comments on the Quality of English LanguageAs a non-native speaker, the manuscript is easy to read and understand. However, there are many grammatical and punctuation errors and in some instances, phrasing need to improve, and poor punctuation in many sentences. Please ask native speakers to help.
Reviewer 2 Report
Comments and Suggestions for Authors
The Authors have prepared a nice manuscript with clear data and figures.
My only (minor) suggestion is to add in the discussion a comparison between the multi-omic approach used by the Authors and other multi-comic approaches in microbiology, specifically on Candida species (e.g., Raman spectroscopy). With adding such a comparison and appropriate references, the paper will become more complete and attract more readers.
Comments on the Quality of English LanguageThe English is reasonably good.
Round 2
Reviewer 1 Report
Comments and Suggestions for Authors
The authors have addressed all my comments and answered the provided questions. Now, the paper has been significantly improved. It can be accepted
Comments on the Quality of English LanguageMinore revision language